# A Comprehensive Characterization of the TI-LGAD Technology

**DOI:** 10.3390/s23136225

**Published:** 2023-07-07

**Authors:** Matias Senger, Anna Macchiolo, Ben Kilminster, Giovanni Paternoster, Matteo Centis Vignali, Giacomo Borghi

**Affiliations:** 1Physics Institute, University of Zurich, Irchel Campus, 8057 Zurich, Switzerland; 2Fondazione Bruno Kessler (FBK), 38123 Trento, Italy; 3Campus Leonardo, Politecnico di Milano, 20133 Milan, Italy

**Keywords:** TI-LGAD, LGAD, 4D-tracking, time resolution, space resolution, 4D-pixels

## Abstract

Pixelated low-gain avalanche diodes (LGADs) can provide both precision spatial and temporal measurements for charged particle detection; however, electrical termination between the pixels yields a no-gain region, such that the active area or *fill factor* is not sufficient for small pixel sizes. Trench-isolated LGADs (TI-LGADs) are a strong candidate for solving the fill-factor problem, as the p-stop termination structure is replaced by isolated trenches etched in the silicon itself. In the TI-LGAD process, the p-stop termination structure, typical of LGADs, is replaced by isolating trenches etched in the silicon itself. This modification substantially reduces the size of the no-gain region, thus enabling the implementation of small pixels with an adequate fill factor value. In this article, a systematic characterization of the TI-RD50 production, the first of its kind entirely dedicated to the TI-LGAD technology, is presented. Designs are ranked according to their measured inter-pixel distance, and the time resolution is compared against the regular LGAD technology.

## 1. Introduction

The trench isolated LGAD (TI-LGAD) technology is one of several sensor technologies aimed at implementing “4D pixels”, which can track ionizing radiation in both space and time. The TI-LGAD is an evolution of the low-gain avalanche diode (LGAD), a silicon radiation detector with an added gain layer for internal signal multiplication [1,2]. This results in reduced active thickness and a remarkable time resolution of tens of picoseconds [3].

In the regular LGAD technology, there is a limit to how small the pixels of a segmented matrix array can be made. This limitation arises from the need for a relatively large termination structure to isolate each pixel, creating a no-gain region of about 50μm [4]. This makes it not convenient for pixel sizes smaller than ∼500μm as the active detector area rapidly decreases. The transversal length of the no-gain region is referred to as interpixel distance (IPD) in this work. This fraction is known as the *fill factor*, which is defined as
$$\mathrm{fill}\;\mathrm{factor}\overset{\mathrm{def}}{=}\frac{\mathrm{active}\;\mathrm{detector}\;\mathrm{area}}{\mathrm{total}\;\mathrm{detector}\;\mathrm{area}},$$
and in a detector with square pixels it can be approximated by
(1)$$\mathrm{fill}\;\mathrm{factor}\;\mathrm{with}\;\mathrm{square}\;\mathrm{pixels}\approx {\left(1-\frac{\mathrm{IPD}}{\mathrm{pitch}}\right)}^{2}$$
where the *pitch* is the pixel cell size. If we consider a pitch of ∼100μm, then the fill factor with standard LGADs is only ∼25%. To overcome this issue, a number of different approaches have been proposed in recent years, such as the resistive silicon detector (RSD) (The AC-coupled resistive silicon detector (AC-RSD), also known as AC-coupled low gain avalanche detector (AC-LGAD) and DC-coupled resistive silicon detector (DC-RSD) are sub families within the RSD technology), the inverse low gain avalanche detector (iLGAD), the deep junction low gain avalanche detector (DJ-LGAD), and the TI-LGAD [5,6,7,8,9,10].

In the TI-LGAD technology, the segmentation of the device into small pixels is achieved by etching trenches in the silicon and filling them with an isolating material [7]. These trenches provide an excellent isolation among neighboring pixels [8] and, as will be discussed in the following sections of this article, reduce the no-gain region to a size that makes it feasible to implement small pixels with a competitive fill factor. Furthermore, as opposed to other alternatives, which require a new and complex manufacturing process, such as DJ-LGADs, or in which the reconstruction algorithms have to be redesigned, such as RSD, the TI-LGAD technology provides a straightforward transition into 4D pixels with manufacturing techniques and algorithms that are already very mature. Additionally, radiation damage processes in TI-LGADs are very similar to those in regular LGAD, which are well-studied already. These reasons make this technology a strong candidate for 4D tracking of ionizing radiation in high energy physics (HEP) experiments.

In this article, we present the results of an in-depth characterization of the TI-LGAD technology. A detailed introduction to the TI-RD50 production and the studied devices is presented in Section 2.1, including details about our irradiation campaign, which can be found in Section 2.1.1. The characterization was performed using three measuring setups, described in detail in Section 2.2. In Section 2.2.1, the transient current technique (TCT) setup is described, which was used to rank the different TI-LGAD designs according to their IPD. In Section 2.2.2, the beta source setup is presented with which the time resolution and collected charge when exposed to a minimum ionizing particle (MIP) was determined. Finally, Section 2.2.3 presents the test beam setup in which similar tests to the beta source setup were performed but with high energy hadrons. Details about the analysis of the data and the used software are presented in Section 2.3. Section 2.3.1 details how the acquired waveforms were processed and the relevant features extracted. Section 2.3.2 details how the IPD is defined in this work, as well as how it is measured. Section 2.3.3 details the calculation of the time resolution and Section 2.3.4 provides details about the measurement of the collected charge. Our findings are presented in Section 3. Section 3.1 details our results for the IPD obtained with the TCT setup. The ranking of the different TI-LGAD designs in terms of their IPD is presented in Figure 12, which is a main result of this work. Section 3.2 discusses the cross-talk and Section 3.3 discusses the time resolution uniformity within a pixel. All of these results were obtained using the TCT setup. Section 3.4.1 details the results of the tests in the beta setup, namely the time resolution and collected charge when the TI-LGADs are exposed to MIP radiation. The results using high-energy hadrons, obtained with the test beam setup, are presented in Section 3.4.2. Finally, Section 4 provides our conclusions for this extensive characterization campaign.

## 2. Materials and Methods

### 2.1. The TI-RD50 Production and Devices Studied

The work presented in this article was carried out on devices manufactured by Fondazione Bruno Kessler (FBK), Trento, Italy, that belong to a production commonly referred to as the *TI-RD50 production* realized in the framework of the RD50 Collaboration [11]. This is the first production of pixelated TI-LGAD and its objective was specifically to provide test devices to evaluate the technology. The TI-RD50 production was composed of a total of 18 wafers, all with an active thickness of 45μm and with different varying characteristics, most of them related to the design of the trenches. These parameters are illustrated with the schematic representations of Figure 1 and described below:**Trench** **depth.** This parameter defines the depth of the trenches that are etched in the silicon. Three different values were explored, namely D1, D2, and D3, with increasing trench depth,
$$\mathrm{trench}\;{\mathrm{depth}}_{D1}<\mathrm{trench}\;{\mathrm{depth}}_{D2}<\mathrm{trench}\;{\mathrm{depth}}_{D3}.$$**Number** **of** **trenches.** This defines the number of trenches that provides isolation between neighboring pixels. In the TI-RD50 production, devices were fabricated either with one or two trenches. In the one-trench designs, a square reticle of trenches was implemented, whilst in the two-trench designs, each pixel was encircled by a trench, thus resulting in two trenches at the interface. The different geometries are illustrated in Figure 1.**Pixel** **border.** This parameter refers to the distance that is left between the edge of the gain layer and the center of the trenches structure. Four values were produced, V1, V2, V3 and V4 with increasing distance,
$$\mathrm{pixel}\;{\mathrm{border}}_{V1}<\cdots <\mathrm{pixel}\;{\mathrm{border}}_{V4},$$
as illustrated in the side view schematic of Figure 1.**Contact** **type.** This defines the way the contact between the external metallization for the interconnection of the pixels and the $n^{++}$ implant is performed. Two different *contact types* were explored, *type 1* and *type 2*, also referred to as *ring* and *dot*, respectively, throughout this work.

The trench depth is a wafer-scale parameter, whilst the number of trenches, pixel border, and contact type can be varied within a single wafer depending on the set of masks used for the fabrication. Devices combining these different parameters values were produced. In this work, devices from wafers 7, 11, and 16 from the TI-RD50 production were studied. These three wafers all share the same doping profiles and differ only by the depth of the trenches, with wafer 7 being medium depth (D2), wafer 11 being shallow depth (D1), and wafer 16 being deep depth (D3). For each depth, i.e., from each wafer, several devices with different numbers of trenches, pixel border, and contact type were studied.

On top of these parameters variations, different layouts comprising pixel sizes as well as matrix sizes (number of pixels) were produced. In this study, those devices with the lowest number of pixels and largest pixel size were selected, as this eases the handling and connection process with discrete readout boards. Due to the large number of possibilities, given all the different parameters combinations, not all designs were available with the same layout. Thus, devices with three different layouts were used:1×2 pixels of $250\times 375\;\mu m^{2}$ size for each pixel.2×2 pixels of $1300\times 1300\;\mu m^{2}$ size for each pixel.4×4 pixels of $250\times 250\;\mu m^{2}$ size for each pixel.

Pictures of these three different layouts are shown in Figure 2. Here, the devices were photographed while mounted on the readout boards and with the respective wire-bonded connections. As shown, in all cases, unused pixels were connected to ground in order to appropriately bias the devices. Only those pixels in use were connected to the output channels of the boards. More details will be provided in the following sections. The different layouts, however, did not play a significant role on the data analysis or the results obtained. A comment will be made wherever the layout of the devices may be relevant.

The devices of the TI-RD50 production were designed to operate, when new and at −20
${}^{\circ }$C, at 200 V.

#### 2.1.1. Irradiation Campaign

Technologies foreseeing applications in HEP experiments must be capable of withstanding high radiation levels. For example, in the CMS Phase-2 inner tracker, radiation levels up to $∼10^{16}\;n_{\mathrm{eq}}\;{\mathrm{cm}}^{-2}$ are encountered [12]. In order to study the resilience of the TI-LGAD technology to radiation damage, an irradiation campaign was carried out. A selected subset of devices was irradiated to different levels at the Jozef Stefan Institute in Ljubljana. The irradiation was performed with reactor neutrons to fluence levels ranging from $1.5\times 10^{15}\;n_{\mathrm{eq}}\;{\mathrm{cm}}^{-2}$ up to $5\times 10^{15}\;n_{\mathrm{eq}}\;{\mathrm{cm}}^{-2}$.

It is worth mentioning that the TI-RD50 production was not intended to withstand high radiation levels as it does not incorporate any mechanism to enhance radiation resilience. A new production is currently underway that incorporates carbon implantation to make the devices more radiation hard.

### 2.2. Experimental Setups

Throughout this work, three different measurement setups were used, a laser TCT setup, a beta source setup, and a test beam setup. Each of these setups is described in detail in the subsections below.

#### 2.2.1. TCT Setup

For a major part of this work, a TCT setup installed in a clean room at the University of Zurich has been used. This TCT setup is a customized unit originally supplied by *Particulars, Advanced Measurement Systems* (More information about the original system can be found in their website https://particulars.si/products.php?prod=LargeScanTCT.html (accessed on 1 July 2023)).

The setup includes a 1064 nm wavelength infrared pulsed laser, which is used to provide the excitation signals for the samples. The intensity of the laser can be adjusted in order to emulate the ionization level of an MIP. This laser is focused by an optical system to produce a spot with a Gaussian profile of (more information can be found in https://msenger.web.cern.ch/a-spacial-characterization-of-the-tct/ (accessed on 1 July 2023)). σ=9±1μm. The samples are attached to a 3D moving stage with ∼1μm precision in x, y, and z. This enables the precise control of the impact position of the laser pulses onto the sample. The electric signals are amplified using 2GHz, 40dB broadband amplifiers and read with a 4GHz, $40\;\mathrm{GS}\;s^{-1}$ oscilloscope.

Our TCT setup incorporates a laser splitting and delaying system to reliably perform time resolution measurements. This system splits the laser pulse in two copies and sends them through paths with different lengths, which produces a 100ns delay between them. A cooling system is installed in the TCT setup to cool the samples down to −20 ${}^{\circ }$C, to facilitate the study of irradiated devices.

#### 2.2.2. Beta Source Setup

An illustration of our beta setup is shown in Figure 3; a picture can be found in Figure 4. The core element of our beta setup is a radioactive ${}^{90}\mathrm{Sr}$ source of beta particles with an activity of 74kBq. The devices are mounted onto a readout board which includes a first amplifying stage. An external second stage amplifier is added to increase the gain. This is a 40–4000MHz low-noise amplifier with a gain of 18dB [13]. The signals are read out by an oscilloscope with the same characteristics as the one in our TCT setup, as described in Section 2.2.1. In order to perform time resolution measurements, a micro-channel plate photon multiplying tube (MCP-PMT) is employed. Although MCP-PMTs are generally intended for single photon detection, they have been shown to provide excellent detection performance with MIPs [14,15,16,17]. The MCP-PMT is connected to the oscilloscope using the same kind of second stage amplifier as for the device under test (DUT). The time resolution of the MCP-PMT was previously determined to be 17±2ps with beta electrons from the ${}^{90}\mathrm{Sr}$ source.

To automate the process of measuring a large quantity of devices, our setup is capable of hosting up to eight DUTs, and is equipped with a robotic system to align both the radioactive source and the MCP-PMT with any of the DUTs.

The setup is placed inside a climatic chamber that enables the temperature to be reduced down to −20 ${}^{\circ }$C, enabling for irradiated samples to be tested.

#### 2.2.3. Test Beam Setup

A subset of devices was brought to a test beam facility at CERN. During these tests, the setup was, in essence, a simplified version of the beta setup from Section 2.2.2 in which the beta source was replaced by a beam of 120GeV pions. Pictures of this setup are shown in Figure 5. The readout boards and second stage amplifiers were the same as in the beta setup, as well as the oscilloscope. In this case, however, the devices were kept at room temperature and the MCP-PMT time reference was not used. The DUTs were stacked along the beam line and measured simultaneously, as shown in the inset of Figure 5. There was no tracking system, i.e., no telescope that enabled us to reconstruct the trajectories of the particles, so that only timing and charge information was collected.

### 2.3. Data Collection and Analysis

In the three setups described in Section 2.2.1, Section 2.2.2 and Section 2.2.3, the data were acquired using the Teledyne LeCroy oscilloscopes as digitizers connected to a computer. The trigger configuration in each setup was different but in all cases the raw waveforms from each trigger and active channel were stored for subsequent offline analysis.

#### 2.3.1. Waveform Analysis

The first analysis performed on each set of data was the analysis of each individual waveform. This analysis was performed in Python running a self-developed analysis package called *signals* and whose source code can be found in https://github.com/SengerM/signals (accessed on 1 July 2023 ). Within this package, the class called *PeakSignal* provides a framework for analyzing peaked signals, as are those produced by TI-LGADs and the MCP-PMT time reference as well. This package automates the extraction of the most relevant features of a signal for the kind of study performed in this work, such as the amplitude, the integral under the peak, the baseline level, the noise, the rise time, etc. An example is shown in Figure 6. Here we see a signal from a TI-LGAD as acquired by an oscilloscope and some of the extracted features. As shown, the software performs a linear interpolation among the samples.

For every measurement performed in this work, each waveform was individually processed with this software and the extracted features were stored in a table-like format file for posterior analysis.

#### 2.3.2. Inter-Pixel Distance

The IPD measures the effective distance among pixels. In a system with completely binary pixels, i.e., pixels, in which we only tag the presence of an impinging particle, the IPD measures the length in between two pixels, in which, if a particle hits, it will not be detected by any of the two neighboring pixels. In the case of LGADs, the IPD is the transversal length of the *no-gain* region in between two pixels. Although a signal may be recorded by the pixels even if a particle impinges in this area, the fact that there is no gain will severely degrade the performance of the detection, thus rendering the device unusable for timing measurements in this condition. The same holds in the case of other technologies, such as the TI-LGAD.

The characteristics of the no-gain region are closely related to the features of the specific technology, and can be hard to determine a priori. The IPD can, however, be empirically measured for each specific device in different ways. In our case, we used the TCT setup described in Section 2.2.1. In this setup, the DUT is excited by fast infrared laser pulses that impinge on a well-defined and accurately controlled position. This makes it possible to study the characteristics as a function of position with a spacial resolution better than 1μm. In order for the infrared photons to reach the silicon, all devices were manufactured with an opening in the top metallization that extends among two neighboring pixels. In Figure 7, a microscope picture of a device is shown. Here, a 1×2 device is shown with the two pixels connected with wire bonds, the third wire bond seen is for the guard ring. The dark brown horizontal rectangle is exposed silicon through an opening in the metallization that spans the two pixels.

In all cases, the laser scans were performed along the dashed line shown in Figure 7, covering a distance larger than the region where the silicon is exposed, i.e., from dot to dot in the drawing. During all the scans, the laser was moved in steps of 1μm and at each specific position multiple events (i.e., laser pulses) were recorded, on the order of 100–500. For each event, the raw waveforms from each pixel were recorded and processed later on, as detailed in Section 2.3.1. In Figure 8, the amplitude measured on each pixel as a function of the position for an example laser scan is shown. Here, the *y* axis displays the normalized amplitude, which is simply the amplitude of the signals, as shown in Figure 6, normalized between 0 and 1. Each color displays the data from each of the two pixels measured, simply called *left* and *right*, and the line in each trace is the average value while the colored bands display the ±1σ of the fluctuations, which, due to the nature of the charge production mechanism with an infrared laser, are Gaussian. The IPD is then computed as the distance from left pixel to right pixel when the amplitude reaches 50% of the average maximum value, by performing a linear interpolation, as shown in the inset in Figure 8. Similar methodologies have been used to measure IPD in the past [1,7].

As above-mentioned, the laser scans were performed spanning a length greater than the region with exposed silicon. There are mainly two reasons for this. In the first place, this enabled us to have a check on the quality of the scan. Since the laser has a Gaussian profile, and each metal–silicon interface is modeled with a *yes–no* transfer function, we expect the amplitude here to follow an erf profile (erf is the *error function* which is the integral of the Gaussian function). Following this simple model, an erf function was fit on each side. An example is shown (only on the left side) in Figure 8. The erf function has two parameters, σ and μ. Here, σ is the size of the laser while μ is the position of the interface between the metal and the exposed silicon. In all cases, σ was used to check that the size of the laser was the one expected (see Section 2.2.1), which may change if the vertical focusing is not correct, while $\mu_{\mathrm{right}}-\mu_{\mathrm{left}}$ was used to calibrate the distance, since we know this value both from the design specification of the devices, as well as from our microscope pictures. The second reason to scan beyond the exposed silicon area is to have data points where there is no signal at all, so then the amplitude normalization between 0 and 1 can be performed. This could, in principle, be performed by using the data from the left pixel when the laser is positioned onto the right pixel and vice versa, however, in the presence of cross-talk this could lead to a bias.

#### 2.3.3. Time Resolution

The time resolution is always determined by using two signal sources, and then the contribution from each is isolated. In the case of the measurements performed in our TCT setup, these two signal sources are the two laser pulses after the optical delay system described in Section 2.2.1. In the case of the beta source setup, the two signal sources are the one coming from the DUT and the one coming from the MCP-PMT, as described in Section 2.2.2. In the case of the measurements at the test beam, the time resolution is computed either among two DUTs or among one DUT and a previously calibrated device.

Independently of which set of signal sources was used to compute the time resolution, in all cases the same software and algorithm was applied. All the analysis was carried out in Python, the source code for the time resolution calculation scripts can be found in https://github.com/SengerM/TI-LGAD_analysis_scripts, https://github.com/SengerM/robocold_beta_setup (accessed on 1 July 2023) and https://github.com/SengerM/220921_test_beam (accessed on 1 July 2023) for measurements from the TCT setup, the beta setup, and the test beam setup, respectively. The different source code from these three repositories is due to the fact that the data formats are different, and so some details had to be adapted; however, the underlying algorithm is the same in the three of them.

To determine the time resolution, a constant fraction discriminator (CFD) algorithm was used, which is a common practice in the field [1]. This algorithm computes the jitter, i.e., the spread in the time difference among two points in two signals, at some constant fraction of the signals. Considering the case in which there are two signals, one coming from a DUT and another coming from the MCP-PMT, for each event there would be two signals, similar to the example signal shown in Figure 6. As shown in this figure, for each signal the *time at x%* is computed with x=10,20,⋯ (see Figure 6). Then we can chose two constant fraction values, e.g., 20% for the DUT and 50% for the MCP-PMT and compute the time difference for each event
$$\Delta t_{20\%,50\%}=t_{\mathrm{DUT}@20\%}-t_{\mathrm{PMT}@50\%}$$

These constant fraction values are usually called CFD values, $k_{\mathrm{CFD}1}$ and $k_{\mathrm{CFD}2}$, etc. The fluctuations of this quantity determine the jitter for this specific condition, and the individual contribution from each $t_{i}$ to these fluctuations is referred to as the time resolution for such a device under these specific conditions. If we assume that the fluctuations arising from each signal are independent, then
(2)$${\mathrm{jitter}}^{2}=\sigma_{1}^{2}+\sigma_{2}^{2},$$
where $\sigma_{i}$ are the time resolutions. An example is shown in the plot on the left in Figure 9. Here, Δt was computed for each event with a fixed value of $k_{\mathrm{CFD}}$. The Δt generally follows a Gaussian distribution, so a Gaussian fit is performed for each case. From this fit, we extract the fluctuations from the σ parameter, which is the jitter. This procedure can be repeated for different values of $k_{\mathrm{CFD}1}$ and $k_{\mathrm{CFD}2}$. In doing this, a two dimensional map of jitter is obtained, as shown in the right chart of Figure 9. From now on the determination of the time resolution can vary depending on the specific case. First of all, a pair of $k_{\mathrm{CFD}1}$ and $k_{\mathrm{CFD}2}$ has to be chosen. There is some freedom for this and ultimately this has to be decided upon, based on some criterion. For example, we can chose those values that minimize the jitter. This would lead to the best possible time resolution that can be achieved with the system. Specific values can also be chosen, and this is especially required when one of the devices was previously calibrated using some specific $k_{\mathrm{CFD}}$ value. In the remainder of the article it will be indicated, wherever necessary, which kind of criterion was used for each case.

Once a specific pair of values of $k_{\mathrm{CFD}}$ was chosen, the procedure to disentangle the time resolution of each device from the measured jitter is also dependent on the conditions in each measurement. For example, if the data were taken in the TCT setup, then the two laser pulses are impinging in the exact same device and position, thus it is very reasonable to consider that in this case $\sigma_{1}=\sigma_{2}$, and so Equation (Equation 2) reduces to $\sigma_{\mathrm{DUT}}=\frac{\mathrm{jitter}}{\sqrt{2}}$. The same holds in the test beam setup if two identical devices are measured simultaneously. In the case of our beta setup, we always have one signal from the DUT and the other signal from the MCP-PMT. Provided that the conditions on the MCP-PMT are kept the same in all the measurements, then $\sigma_{\mathrm{MCP}-\mathrm{PMT}}$ is constant and $\sigma_{\mathrm{DUT}}=\sqrt{{\mathrm{jitter}}^{2}-\sigma_{\mathrm{MCP}-\mathrm{PMT}}^{2}}$, where $\sigma_{\mathrm{MCP}-\mathrm{PMT}}$ was determined beforehand via a calibration procedure (see Section 2.2.2). The same holds for the cases in which a DUT is measured against a previously calibrated device in the test beam setup.

#### 2.3.4. Collected Charge

The collected charge refers to the charge that is collected at the output electrodes by the readout electronics after a particle impinged on the DUT. This is an important parameter since there are minimum thresholds required to ensure proper operation. Since the amplifiers used in this work behave as transimpedance amplifiers, the charge is proportional to the time integral of the signals. As described in Section 2.3.1, the integral under the peak was calculated for every signal acquired, as shown in the example of Figure 6. It is a known fact that the charge deposited by an MIP in a silicon detector follows a Landau distribution [1,18]. To account for the noise contribution, the so called *Langauss* distribution is used, which is a Landau convoluted with a Gaussian. Thus, a Langauss function was fitted to each dataset to determine the collected charge. An example is shown in Figure 10. In this plot, we see the distribution of the collected charge for an example measurement from our beta setup. On top of the histogram, the Langauss fit is shown together with the Landau component. From this fit, the collected charge is determined to be the most probable value of the underlying Landau component, which in this example is around 22pVs.

## 3. Results

In this section, the results from each of the studies performed will be presented and discussed.

### 3.1. Inter-Pixel Distance

As detailed in Section 2.3.2, the IPD provides a measure of the size of the no-gain region. In order to maximize the fill factor, the IPD is as small as possible. Thus, we will rank the different designs that were tested according to this criterion.

Figure 11 shows the measured IPD as a function of the bias voltage. In this plot, each single point is an IPD measurement, as described in Section 2.3.2, while each trace is a different tested device. The left chart shows devices with 1 trench and the right chart devices with 2 trenches. The color denotes the pixel border, the line dash the contact type, and the marker shape is the trench depth. All these parameters were introduced in Section 2.1. We note that the IPD is a strong function of these design parameters. We also note that all designs have an improved IPD with respect to the typical plain-LGAD technology.

For some of the designs, such as 1 trench, V1, ring and D1, an IPD of 0μm (within the measurement uncertainty) is obtained at the working voltage. This makes the TI-LGAD technology very promising for solving the fill-factor issue. For some designs a negative IPD is observed above the working voltage. This phenomenon arises due to additional multiplication in the inter-pixel region due to the trench structure. Though this may look as an enhanced feature of the technology, in this regime the devices have a high auto-trigger rate which, depending on the application, may render them unusable. This means that the sensor produces signals that look like particles impinging in the pixel but are not due to actual particles, leading to fake events.

In Figure 12, a chart is shown ranking the different designs according to the measured IPD at working voltage. In this chart, we find several designs with an IPD smaller than 2μm, which leads to fill factor values higher than 95% for a pitch of 100μm or higher than 92% for a pitch of 50μm using Equation (Equation 1).

After irradiation, all designs seem to have the same IPD of about 2–4μm independent of their parameters and of the bias voltage as well. This seems to be a beneficial effect since a lower IPD is always desired. This points to the fact that all the structure we find in the plots of Figure 11 and Figure 12 comes from an interplay between the design of the trenches and the gain layer. After the devices have been irradiated, the gain layer becomes less active, as is usual in LGADs [1], and so the differences in the IPD become less evident. In any case, no adverse effects were observed after irradiation up to $2.5\times 10^{15}\;n_{\mathrm{eq}}\;{\mathrm{cm}}^{-2}$.

### 3.2. Cross Talk and Radiation Resilience of Trenches

When a particle impinges on some pixel, it may happen that some signal leaks to neighboring pixels. This is known as *cross-talk*. There are different mechanisms that can lead to a cross-talk among the pixels. In the case of TI-LGADs, one may wonder whether the trenches provide an adequate isolation between the pixels. According to our studies, the trenches not only provide an excellent isolation mechanism with negligible cross-talk when devices are new, but they also withstand, with no perceptible degradation, the radiation levels up to at least a fluence of $2.5\times 10^{15}\;n_{\mathrm{eq}}\;{\mathrm{cm}}^{-2}$. Higher fluences were not studied due to the severe degradation of the charge collection efficiency. A systematic study was not performed because the cross-talk was, in all cases, below the noise floor.

### 3.3. Time Resolution Uniformity

Another important study performed with data gathered from our TCT setup, described in Section 2.2.1, is the uniformity of the time resolution along the pixels. For this study, we used the two laser pulses provided by the delay system. An example of the time resolution uniformity measurement is shown in Figure 13. In this plot, each point is a time resolution measurement as described in Section 2.3.3, and is repeated at different positions for each of the two measured pixels. Since the measurement was performed with a laser instead of a charged particle, the absolute value of time resolution is not relevant as it misses the Landau noise. However, we can study how the time resolution changes as a function of the position. We see that it is very uniform along the pixels until very close to the edges. Moreover, we only find deviations when the laser is closer to the edge than its own size, as indicated in Figure 13. We conclude that for the scan shown in this figure the time resolution uniformity within the pixel is almost perfect. Though this is an example from a single measurement, the same behavior was observed in all tested devices. A more detailed and systematic study was not carried out since the non-uniformities are smaller than the measuring capabilities of our setup and, since they are so small, it does not make sense for our interests.

### 3.4. Time Resolution and Collected Charge

In this section, the results regarding time resolution and collected charge from our beta setup and test beam setup will be discussed. These setups, respectively, are introduced in Section 2.2.2 and Section 2.2.3, and provide the most reliable and realistic way of measuring these parameters since the DUT is excited by real charged particles, as opposed to laser pulses in the TCT setup.

#### 3.4.1. Beta Setup

The plots in Figure 14 summarize our results with the beta setup. The plot on the left shows the time resolution as a function of the bias voltage, while the plot on the right is the collected charge as a function of the bias voltage. In the left plot, each point is a time resolution measurement as described in Section 2.3.3, and in the right plot each point is a charge measurement as defined in Section 2.3.4. Each trace in these plots is a different device, and the color encodes the irradiation fluence. As seen in the plots, the parameters related to the trenches that were introduced in Section 2.1 are not encoded in any way in these plots. The reason is that we did not observe any systematic dependence of the time resolution or the collected charge on them, which is expected. For the non-irradiated devices, we see that the behavior is close to optimal obtaining time resolution values lower than ∼30ps and charge values up to ∼30fC. In calculating the time resolution, the value of $k_{\mathrm{CFD}}$ for the MCP-PMT was kept fixed at 20% as per calibration requirement, while for the DUTs it was also kept at fixed values but different for each fluence value:$k_{\mathrm{CFD}}=20\%$ for $0\times 10^{15}\;n_{\mathrm{eq}}\;{\mathrm{cm}}^{-2}$;$k_{\mathrm{CFD}}=40\%$ for $1.5\times 10^{15}\;n_{\mathrm{eq}}\;{\mathrm{cm}}^{-2}$;$k_{\mathrm{CFD}}=50\%$ for $2.5\times 10^{15}\;n_{\mathrm{eq}}\;{\mathrm{cm}}^{-2}$.

This was performed after finding that a better time resolution can be obtained by changing this parameter as a function of the fluence. The reason for this is related to the degradation of the gain layer; as the signals become smaller they “sink” into the noise floor and so a higher fraction of the amplitude is required to obtain a clean rising edge. This does not pose any bias towards a real application of the technology as changing this parameter (or a similar one) as a function of the fluence is easy and completely feasible.

In the case of irradiated devices we see that the required voltage values are considerably higher. Also, the time resolution and the collected charge are degraded due to the deactivation of the gain layer as a consequence of radiation damage. These effects are not surprising, indeed expected, as they are common to all LGAD technologies with no additional processing, as for example the co-implantation of carbon in the boron gain layer, is carried out. We do not have a clear explanation for the larger spread in the results with irradiated devices, especially in the collected charge. However, we suspect these are device-specific differences, and not related to the design of the trenches as no systematic behavior could be found.

With respect to the apparently improved time resolution obtained with $2.5\times 10^{15}\;n_{\mathrm{eq}}\;{\mathrm{cm}}^{-2}$ in comparison with the $1.5\times 10^{15}\;n_{\mathrm{eq}}\;{\mathrm{cm}}^{-2}$ at voltages lower than ∼400V, there is a hidden efficiency loss accounting for this. To understand this, it is helpful to look at the plots in Figure 15 where we show how the event selection was performed for three particular examples, one at each fluence. In these plots each dot is one event in the plane amplitude vs. peak time, which are the waveform features, as shown in Figure 6. For the example at $0\;n_{\mathrm{eq}}\;{\mathrm{cm}}^{-2}$ (i.e., non-irradiated), the separation into signal and background events can simply be achieved by a threshold set for the amplitude, since the gain is enough to separate the signal events from the background. As the irradiation fluence increases, the gain becomes suppressed due to the radiation damage in the multiplication layer. As a consequence, the amplitude of the signals is no longer so high that the signal events can be trivially distinguished from the background events arising from the noise inherent in the system. This is particularly evident in the rightmost plot of Figure 15 where, in the inset, it can be seen that the population of signal events “sinks” into the background events. This is directly translated into a hit efficiency loss (i.e., a fraction of events that are not detected), and an improved time resolution, since now we are only taking into account those signals in the higher end of the Landau distribution. In the case of the $1.5\times 10^{15}\;n_{\mathrm{eq}}\;{\mathrm{cm}}^{-2}$ (middle plot) we can still obtain a reasonable hit efficiency. Unfortunately, it is very hard to obtain a reliable hit efficiency estimation with the beta setup, so we cannot quantify this effect.

As it is common practice in the field, in Figure 16 the time resolution and collected charge data are presented in a single plot as a function of one another. The data points shown in the top plot are the same as those from Figure 14. In the bottom plot, an average over the fluence is shown.

#### 3.4.2. Test Beam

The studies performed in the test beam are, in essence, the same as in our beta setup, but with 120GeV pions instead of beta particles. Due to technical limitations, the devices were kept at room temperature and so only non-irradiated samples could be tested. The main impact of temperature is on the required bias voltage to operate the sensors, which must be higher at higher the temperatures. It also has a small impact on the noise, slightly degrading the time resolution. As detailed in Section 2.2.3, the setup had no tracking capabilities and so only charge and time resolution studies were performed. The time resolution and collected charge as a function of bias voltage can be found in Figure 17. In the left plot, each point is a time resolution measurement as described in Section 2.3.3, and in the right plot each point is a charge measurement as defined in Section 2.3.4. Each color is a different device and each trace is a different voltage scan, in some cases more than one voltage scan were repeated for the same device. The marker is denoting the pixel size, which is indicated only due to the fact that devices with small pixels accumulate lower statistics.

Both the time resolution and the collected charge seem to be in agreement with the results obtained with the beta setup. The comparison between test beam and beta setup measurements is easier when looking at the time resolution vs. collected charge plot shown in Figure 18. When comparing this plot (Figure 18) with the one for the beta setup (Figure 14) we see that the results are in agreement. The time resolution is slightly worse in the test beam setup, but the higher temperature accounts for this difference.

## 4. Conclusions

A systematic and in-depth characterization of the TI-RD50 production was presented. This production is the first one exclusively dedicated to the TI-LGAD pixel technology and was intended for exploring different designs related to the trenches mechanism and the optimization of trench fabrication parameters.

Different TI-LGAD samples were ranked according to the size of the no-gain region between the pixels, which was empirically measured with an infrared laser TCT setup. This is probably the most important parameter to qualify the TI-RD50 production, as the goal of the TI-LGAD technology is to solve the *fill factor issue* that limits the application of the regular LGAD technology. A number of TI-LGAD designs were found with IPD lower than 2μm, this is approximately 10 to 50 times smaller than regular LGADs. The three designs with the lowest IPD are from wafer 16, trench depth D3, contact type ring, pixel border versions V1 and V2, and 1 and 2 trenches. The complete details were presented in Figure 12.

The time resolution and collected charge were measured, both with beta particles and with high energy pions. In both cases, the results are in agreement with the outstanding characteristics of the LGAD technology, providing a time resolution on the order of 30ps. Charge and time resolution were found out to be independent of the trenches design.

The time resolution as a function of the position was also studied using the infrared laser TCT setup. Our findings show that the time resolution is perfectly uniform within the pixel area up to at least 9μm from the pixel edge, which is the limit for our measuring setup given by the laser size.

These characteristics put the TI-LGAD technology as a strong candidate towards the implementation of 4D pixels. Moreover, a number of samples were irradiated to fluences comparable to those found in HEP detectors environments. Although the TI-RD50 production does not incorporate any mechanism to improve the radiation hardness of the devices, we verified that they are still operational up to a fluence of $1.5\times 10^{15}\;n_{\mathrm{eq}}\;{\mathrm{cm}}^{-2}$. At higher values of fluence we observe a severe reduction in the collected charge related to the degradation of the gain mechanism, but the isolation provided by the trenches, as well as the IPD are not degraded. It remains a task for future productions to increase the radiation hardness of the technology. One such productions is currently underway and we expect results in the near future.

## Figures and Tables

**Figure 1 sensors-23-06225-f001:**
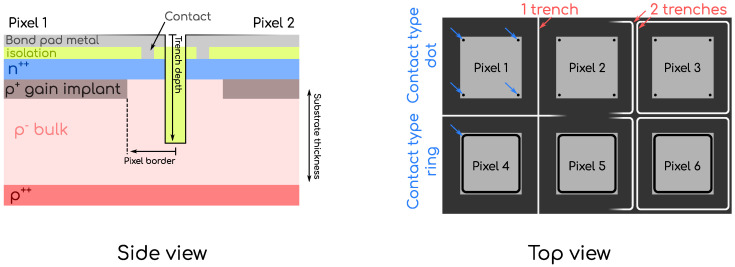
Schematic representation of the different design parameters that were explored in the TI-RD50 production. The left panel shows a detailed side view of the trench area, and illustrates the *pixel border*, as well as the *trench depth*. The right panel displays a top view illustrating the *number of trenches* and the *contact type*.

**Figure 2 sensors-23-06225-f002:**
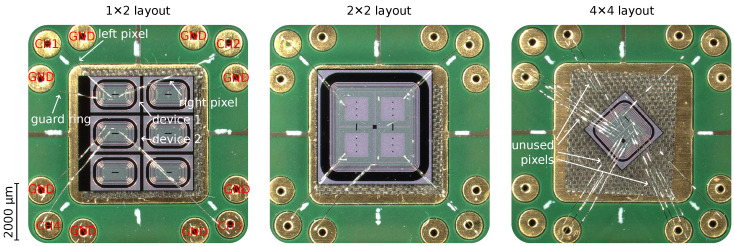
Pictures of three devices mounted on the readout boards showing the three different layouts used during this work.

**Figure 3 sensors-23-06225-f003:**
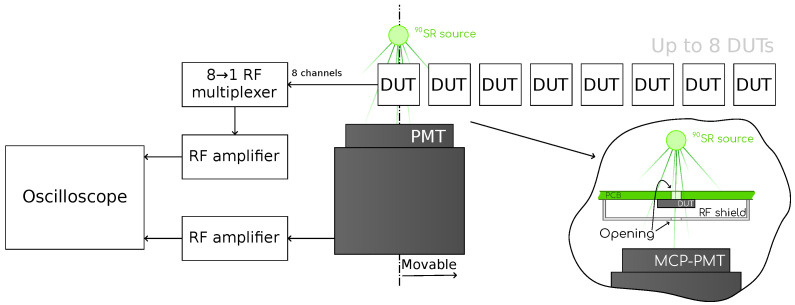
Schematic representation of the beta setup. A picture of this setup is shown in Figure 4.

**Figure 4 sensors-23-06225-f004:**
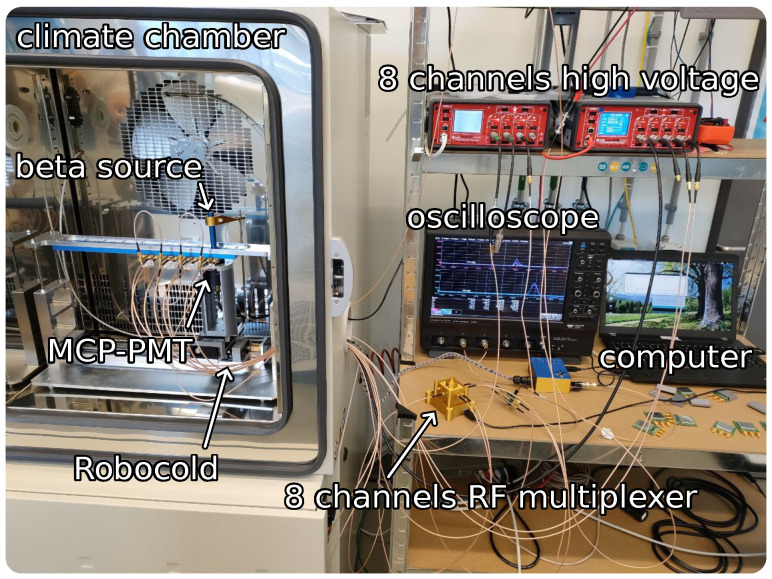
Picture of the beta setup utilized in this work. A schematic representation of this setup is shown in Figure 3.

**Figure 5 sensors-23-06225-f005:**
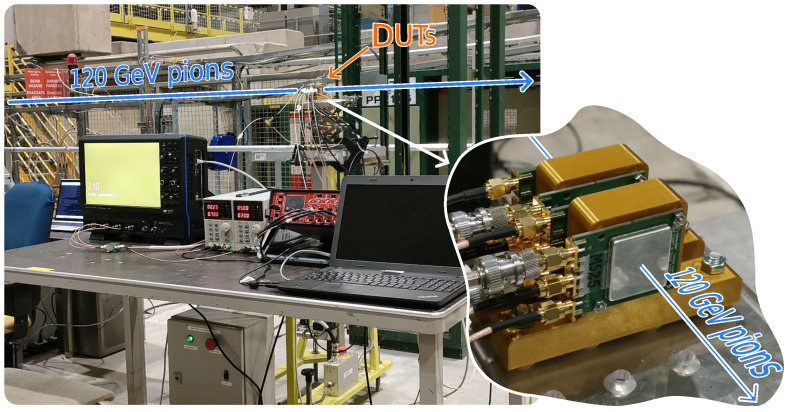
Pictures of the setup at the test beam at CERN.

**Figure 6 sensors-23-06225-f006:**
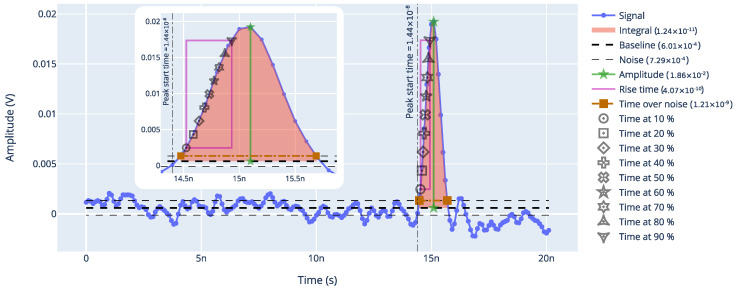
Example of an LGAD signal analyzed with the developed software.

**Figure 7 sensors-23-06225-f007:**
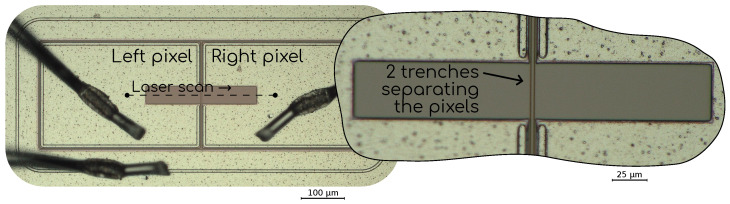
Microscope picture of a 1×2 device, showing the line along which the laser scan was performed in the TCT setup. The wire bonds can be seen landing on the metallization of the left and right pixels, respectively, as well as on the guard ring. The darker brown region is exposed silicon with no metallization through which the infrared laser was shined onto the sample. The inset shows a detail of the inter-pixel region and the two darker vertical lines that go through the metallization opening are the trenches, in this case the device has two trenches.

**Figure 8 sensors-23-06225-f008:**
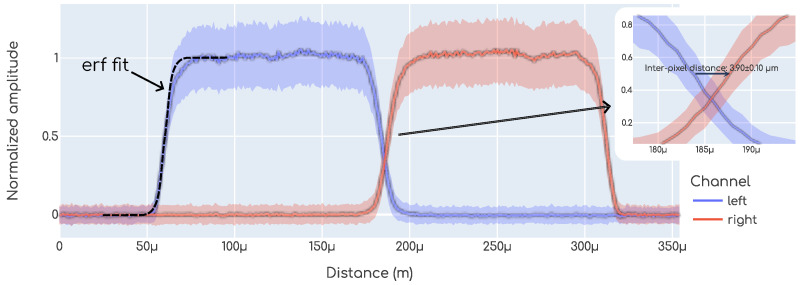
Example of an IPD measurement.

**Figure 9 sensors-23-06225-f009:**
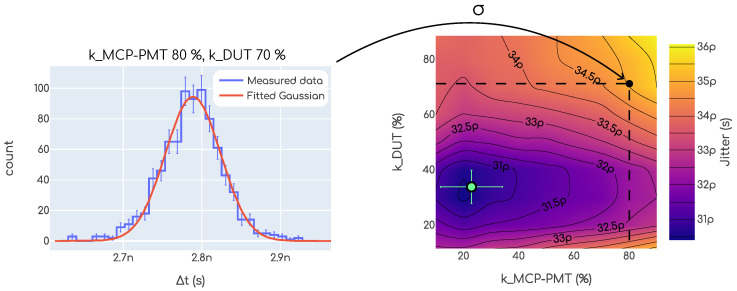
Jitter calculation example for the determination of the time resolution. This example is from our beta setup. Similar plots are obtained for a single position in the TCT setup, as well as in the test beam setup.

**Figure 10 sensors-23-06225-f010:**
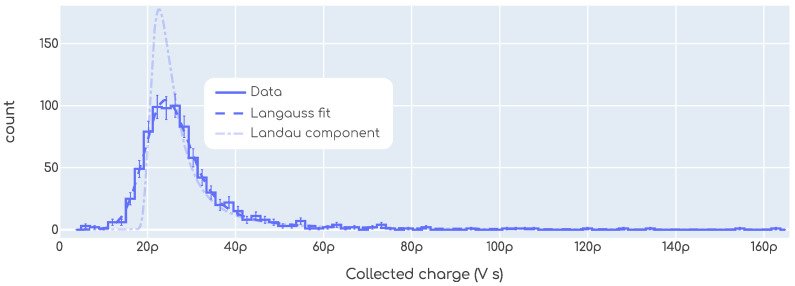
An example showing the Landau distribution of the collected charge in our beta setup. A *Langauss* fit is shown which is a Landau convoluted with a Gaussian. The Landau component is also plotted. The units of charge are before the conversion to Coulomb, i.e., before dividing by the transimpedance of the system.

**Figure 11 sensors-23-06225-f011:**
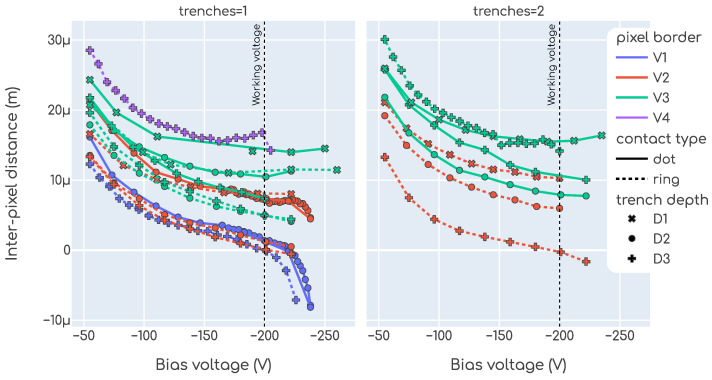
Inter-pixel distance as a function of the applied bias voltage for all the designs tested (non-irradiated devices).

**Figure 12 sensors-23-06225-f012:**
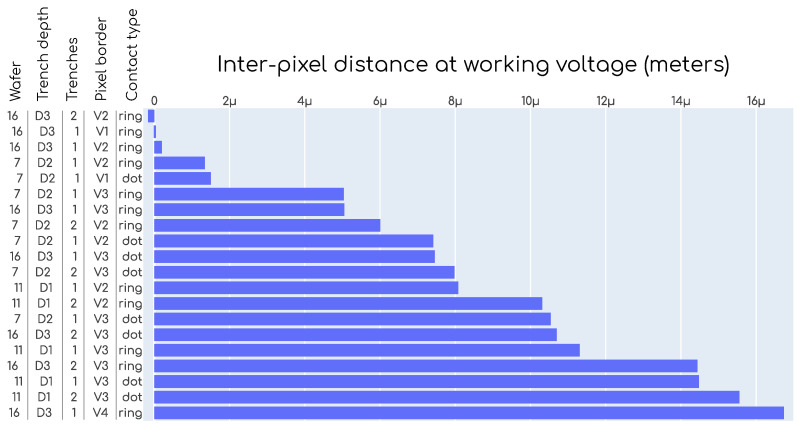
TI-LGAD designs ranked according to increasing IPD (non-irradiated devices). These values were obtained at −20 deg Celsius and with a bias voltage of −200V (see Figure 11, the vertical line labeled *working voltage)*.

**Figure 13 sensors-23-06225-f013:**
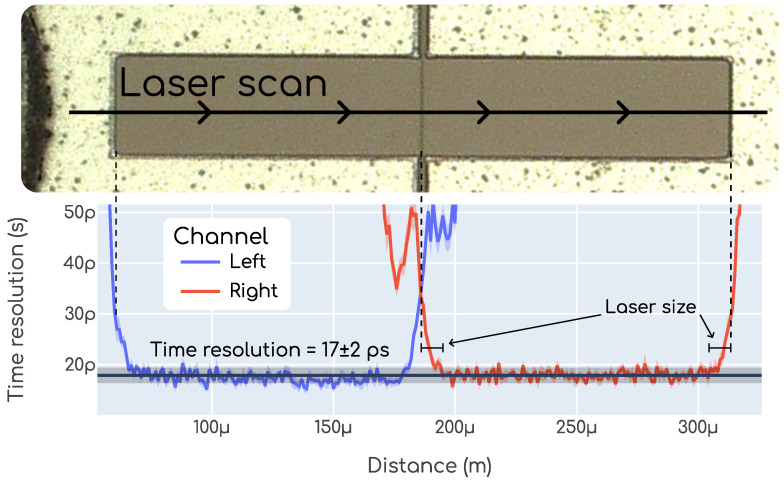
Time resolution uniformity for a specific scan with the TCT setup.

**Figure 14 sensors-23-06225-f014:**
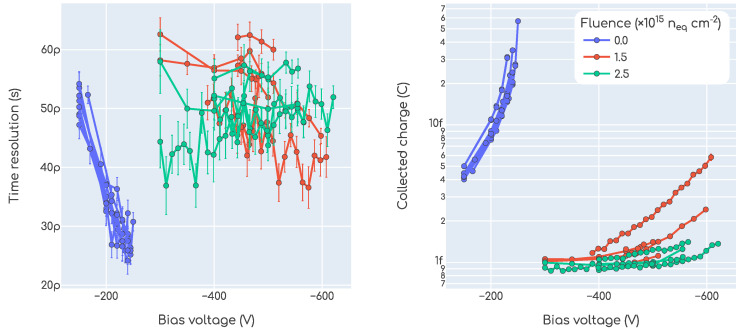
Time resolution and collected charge as a function of the bias voltage measured with the beta setup.

**Figure 15 sensors-23-06225-f015:**
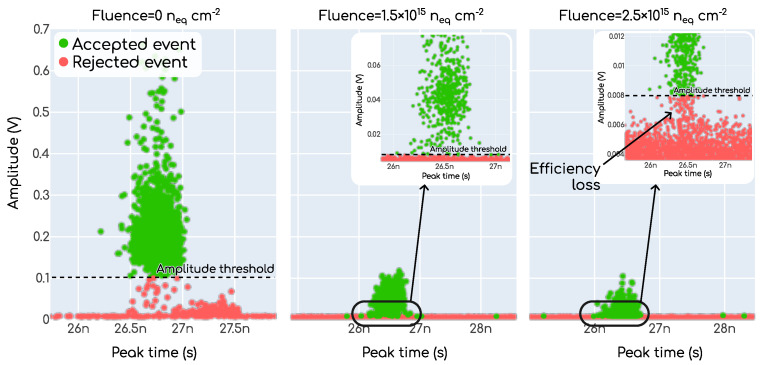
Distribution of events in an amplitude vs. peak time plane showing how the event selection with a threshold on the amplitude leads to no efficiency loss with non-irradiated devices but not in the case of irradiated devices.

**Figure 16 sensors-23-06225-f016:**
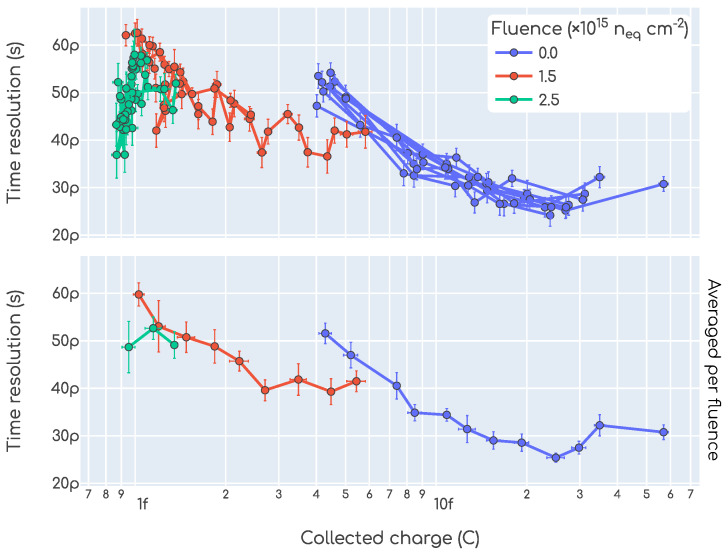
Time resolution as a function of collected charge obtained in our beta setup.

**Figure 17 sensors-23-06225-f017:**
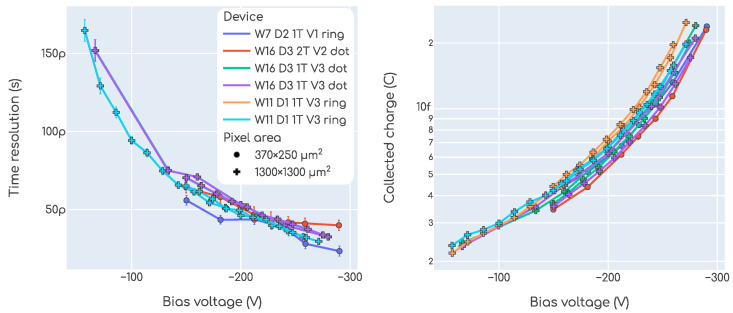
Time resolution and collected charge as function of the bias voltage measured with the test beam setup.

**Figure 18 sensors-23-06225-f018:**
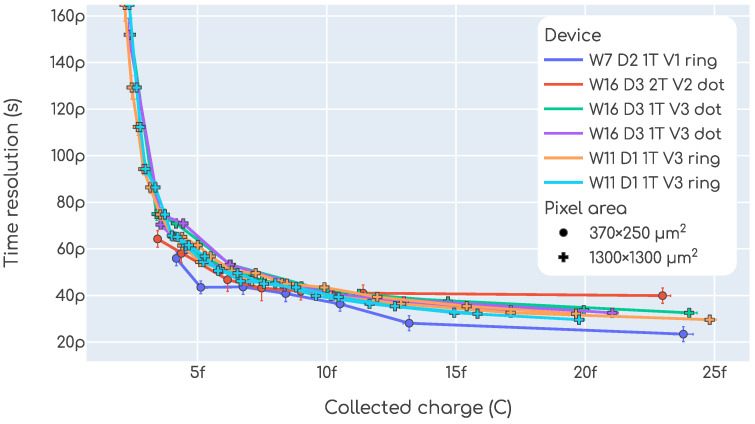
Time resolution as a function of collected charge obtained in the test beam setup.

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
