# Peer review of "A Comprehensive Characterization of the TI-LGAD Technology"

_sensors, 2023, doi:10.3390/s23136225_

Round 1
Reviewer 1 Report
The work of this study is good, however, English should be improved. Also a lot of missing words are found. I write down some examples
<<< correct this section title 2.1. The ????and devices studied
<<<line 84 The ؟؟؟؟ was composed of a total of 18 wafers,
<<<page 4 , lin 87 please repair this stament " It is worth mentioning that the was not intended to withstand high radiation levels.........
<<<page 10 front of pag, this "......for each event." should moved before the equation
<<<page 20, line 439 put "," afer Unfortunately
<<<in conclusion section
A systematic an in depth characterization of the?????? was presented.
also correct "high energy pions. "
Although the ???? does not
English should be improved. Also a lot of missing words are found. I write down some examples
<<< correct this section title 2.1. The ????and devices studied
<<<line 84 The ؟؟؟؟ was composed of a total of 18 wafers,
<<<page 4 , lin 87 please repair this stament " It is worth mentioning that the was not intended to withstand high radiation levels.........
<<<page 10 front of pag, this "......for each event." should moved before the equation
<<<page 20, line 439 put "," afer Unfortunately
<<<in conclusion section
A systematic an in depth characterization of the?????? was presented.
also correct "high energy pions. "
Although the ???? does not
Author Response
Dear reviewer. Thank you for your comments and suggestions. All your comments were addressed and were fixed in the newer version.
✅ <<< correct this section title 2.1. The ????and devices studied
✅ <<<line 84 The ؟؟؟؟ was composed of a total of 18 wafers,
✅ <<<page 4 , lin 87 please repair this stament " It is worth mentioning that the was not intended to withstand high radiation levels.........
✅ <<<page 10 front of pag, this "......for each event." should moved before the equation
✅ <<<page 20, line 439 put "," afer Unfortunately
<<<in conclusion section
✅ A systematic an in depth characterization of the?????? was presented.
✅ I don't understand it: also correct "high energy pions. "
The paragraph was improved.
✅ Although the ???? does not
Reviewer 2 Report
m-minor comment, M - major, followed by line number.
m 22 - do you need all 5-10 references? just a suggestion to cut the oldest ones out.
m 36 - The ?? and devices - a missing word?
m 43 - rephrase? These parameters are illustrated with the schematic representations in figure 1? otherwise it sounds like figure 1 is schematically represented..
m82, For example, (comma is missing)
m85 - Jožef Stefan or Josef? JoÅŸef Stefan - this is how it looks on my PC
m87 - worth mentioning that the ??? was not intended-missing word
m126 - can produce up to 1 kV? can output? using word 'source' is a bit unusual here judging from US English perspective but i don't have a strong opinion here :) as the meaning is clear
(just a future thought, CAEN makes very nice stand-alone flash ADCs that work more reliable than a scope for data, no need for a crate.. I've been using a scope before but when i got some funding and switched to ADC DT5730 / DT5730S its much much faster too, like 800 triggers/second, but its almost 10k $.. i'm unrelated to caen, this is not an ad in any way, just personal experience with a similar setup to your beta setup. used them for the beam tests too.)
(thought 2 - looked at your python code.. quick suggestion - i normally use ROOT with python (PyROOT) and its rather faster than of pure python code, as ROOT provides pre-compiled binaries that do work faster.. although numpy is based on the similar idea. But this was you can use the landau and similar fits without doing the porting and at a faster speed).
m Figure 12 - the figure is too dense with information.. may be split it into 2 or more? or give it a better caption?
m364 - withstand, with no perceptible degradation, the .... add commas or rephrase.
Author Response
Dear reviewer. Thank you for your comments and suggestions. Below I address each of your comments individually. The icon '✅' means we have implemented your comment/suggestion; the icon '❌' means we didn't, in which case we provide a justification of why we would like to keep without such change.
❌ m 22 - do you need all 5-10 references? just a suggestion to cut the oldest ones out.
Thanks for pointing this out. However, if it is fine for you we would prefer to keep them for future reference.
✅ m 36 - The ?? and devices - a missing word?
✅ m 43 - rephrase? These parameters are illustrated with the schematic representations in figure 1? otherwise it sounds like figure 1 is schematically represented..
✅ m82, For example, (comma is missing)
✅ m85 - Jožef Stefan or Josef? JoÅŸef Stefan - this is how it looks on my PC
✅ m87 - worth mentioning that the ??? was not intended-missing word
✅ m126 - can produce up to 1 kV? can output? using word 'source' is a bit unusual here judging from US English perspective but i don't have a strong opinion here :) as the meaning is clear
✅ (just a future thought, CAEN makes very nice stand-alone flash ADCs that work more reliable than a scope for data, no need for a crate.. I've been using a scope before but when i got some funding and switched to ADC DT5730 / DT5730S its much much faster too, like 800 triggers/second, but its almost 10k $.. i'm unrelated to caen, this is not an ad in any way, just personal experience with a similar setup to your beta setup. used them for the beam tests too.)
Thank you for the suggestion. Actually, we recently acquired a CAEN DT5742 which is similar to an oscilloscope in performance (actually slightly worse) but has 16 channels. It is probably similar to yours, from what we have found in the website. About the trigger rate, if your oscilloscopes support "fast mode" you can actually get much higher trigger rates. As an example, with [our oscilloscope](https://www.teledynelecroy.com/oscilloscope/waverunner-9000-oscilloscopes/waverunner-9254m) it is possible to get 1e6 triggers/second (look in the 'specs' tab and search for 'Maximum Trigger Rate').
✅ (thought 2 - looked at your python code.. quick suggestion - i normally use ROOT with python (PyROOT) and its rather faster than of pure python code, as ROOT provides pre-compiled binaries that do work faster.. although numpy is based on the similar idea. But this was you can use the landau and similar fits without doing the porting and at a faster speed).
Thank you for the suggestion, we will take it into account for future analyses.
❌ m Figure 12 - the figure is too dense with information.. may be split it into 2 or more? or give it a better caption?
Thank you for your comment on this, we acknowledge that it is not a regular kind of plot/table. After discussing this among the authors, we think it is fine and would like to keep it like this, if it is fine for you. This figure presents, in essence, a table in which the last column was plotted. An alternative would be a regular table with numeric entries, but we consider that in the current way it makes it easier for the reader to see "how good" is each design compared with the others.
✅ m364 - withstand, with no perceptible degradation, the .... add commas or rephrase.
Reviewer 3 Report
The manuscript presents a systematic and comprehensive characterization of the TI-LGAD technology, focusing on the reduction of the no-gain region and the inter-pixel distance. The results show promising advancements in solving the fill factor issue of LGAD technology, with significant reductions in the inter-pixel distance and consistent time resolution performance.
Overall, the manuscript provides valuable insights into the TI-LGAD technology, highlighting its potential for improving the fill factor and maintaining excellent time resolution. The characterization efforts and findings presented contribute to advancing the understanding and applicability of TI-LGAD technology in pixel-based applications. The manuscript can be recommended for publication in Sensors, after the authors addressing the following major questions.
1. How does the negative inter-pixel distance (IPD) observed above the working voltage impact the usability of TI-LGAD devices?
2. What is the conclusion regarding the uniformity of time resolution along the pixels based on the study mentioned in figure 13?
3. The manuscript could benefit from enhanced conciseness and improved clarity in presenting the data. The verbosity of the text and the lack of clear data representation hinder the effectiveness of conveying the research findings.
4. The introduction section is relatively brief and lacks a thorough review of the relevant research work.
5. The entire manuscript contains numerous grammar errors that require thorough revision.
6. The description of the setup parameters on page 5 is excessively verbose and lengthy. It needs to be condensed.
The entire manuscript contains numerous grammar errors that require thorough revision.
Author Response
Dear reviewer. Thank you for your comments and suggestions. Below I address each of your comments individually.
✅ 1. How does the negative inter-pixel distance (IPD) observed above the working voltage impact the usability of TI-LGAD devices?
Thanks for pointing this up. Up to the working voltage it has no effect in their usability, because this effect arises for higher voltages. Above the working voltage this effect could be exploited as long as the hit rate is higher than the dark rate that gives rise to this effect. This would, however, make the device less uniform throughout its surface. We include a discussion of this in the paragraph just below figure 11.
✅ 2. What is the conclusion regarding the uniformity of time resolution along the pixels based on the study mentioned in figure 13?
We included a paragraph about this topic to the conclusions, thanks for mentioning it.
✅ 3. The manuscript could benefit from enhanced conciseness and improved clarity in presenting the data. The verbosity of the text and the lack of clear data representation hinder the effectiveness of conveying the research findings.
Thanks for this comment. We checked some similar papers and find that our presentation of the time resolution and charge results are presented in a standard fashion. The inter-pixel distance for each design is presented in a table in which the inter-pixel distance column was replaced by a bar chart to make it easier for the reader to visualize how "good" is each design. We hope that this way of presenting the results is more clear.
✅ 4. The introduction section is relatively brief and lacks a thorough review of the relevant research work.
A new paragraph with a review of the research work has been added to the introduction section.
✅ 5. The entire manuscript contains numerous grammar errors that require thorough revision.
We have revised the grammar, and apologize for the mistakes in the original version.
✅ 6. The description of the setup parameters on page 5 is excessively verbose and lengthy. It needs to be condensed.
Indeed, it was too lengthy on a second thought. We have simplified the description of the setup parameters and configuration.
Round 2
Reviewer 3 Report
The manuscript can be accepted in present form.